# Effects of the Quinone Oxidoreductase WrbA on *Escherichia coli* Biofilm Formation and Oxidative Stress

**DOI:** 10.3390/antiox10060919

**Published:** 2021-06-06

**Authors:** Federico Rossi, Cristina Cattò, Gianmarco Mugnai, Federica Villa, Fabio Forlani

**Affiliations:** Department of Food, Environmental and Nutritional Sciences, Università degli Studi di Milano, 20133 Milano, Italy; federicorossi1979@gmail.com (F.R.); cristina.catto@unimi.it (C.C.); gianmarco.mugnai.87@gmail.com (G.M.); fabio.forlani@unimi.it (F.F.)

**Keywords:** WrbA, quinone oxidoreductase activity, biofilm formation, oxidative stress, mature biofilm, *Escherichia coli*

## Abstract

The effects of natural compounds on biofilm formation have been extensively studied, with the goal of identifying biofilm formation antagonists at sub-lethal concentrations. Salicylic and cinnamic acids are some examples of these compounds that interact with the quinone oxidoreductase WrbA, a potential biofilm modulator and an antibiofilm compound biomarker. However, WrbA’s role in biofilm development is still poorly understood. To investigate the key roles of WrbA in biofilm maturation and oxidative stress, *Escherichia coli* wild-type and ∆*wrb*A mutant strains were used. Furthermore, we reported the functional validation of WrbA as a molecular target of salicylic and cinnamic acids. The lack of WrbA did not impair planktonic growth, but rather affected the biofilm formation through a mechanism that depends on reactive oxygen species (ROS). The loss of WrbA function resulted in an ROS-sensitive phenotype that showed reductions in biofilm-dwelling cells, biofilm thickness, matrix polysaccharide content, and H_2_O_2_ tolerance. Endogenous oxidative events in the mutant strain generated a stressful condition to which the bacterium responded by increasing the catalase activity to compensate for the lack of WrbA. Cinnamic and salicylic acids inhibited the quinone oxidoreductase activity of purified recombinant WrbA. The effects of these antibiofilm molecules on WrbA function was proven for the first time.

## 1. Introduction

Biofilm growth characterizes microbial life in natural and engineered ecosystems [1]. The acquisition of a biofilm lifestyle allows cells to cope with unfavorable environmental conditions and to colonize new niches [2,3,4]. In the biofilm arrangement, the cells differ significantly from their planktonic counterparts in their functional and biochemical properties, as well as in their expression of specific genotypic and phenotypic traits [5,6]. Biofilm-dwelling cells increase their tolerance to biotic and abiotic stresses, such as predation, ultraviolet radiation, harsh temperatures, drought, salinity, and oxidative stress. Biofilms can be found on virtually every type of biotic or abiotic surface due to their plasticity [1]. However, some biofilm types are a threat to humans [7,8,9], animals [10,11], plants [12,13], and man-made systems [14,15,16], such that the acquisition of multicellular lifestyles and the underlying mechanisms has become a field of investigation. The control treatments that remove the unwanted biofilms and the substrate injuries the biofilms inflict represent major socioeconomic burdens in our modern society. The most detrimental characteristic of biofilms is the expression of specific genes that make the sessile cells up to 1000-fold more resistant to antimicrobial agents than their planktonic counterparts [17,18].

In light of the above considerations, the scientific community is painstakingly scrutinizing the mechanisms behind the genesis of these biofilms, aiming to find their Achille’s heel to incapacitate the microorganisms without killing them. This novel strategy has the advantage of reducing biocides that are harmful to humans and the environment and reducing the spread of antimicrobial resistance [19].

The effects of natural compounds on biofilm formation have been extensively studied, with the goal of identifying the biofilm formation antagonists at sub-lethal concentrations [20,21,22,23]. In fact, the natural compounds at sub-lethal concentrations can interfere with the key steps in biofilm genesis, namely attachment, maturation, and detachment, discouraging the sessile mode of microorganisms [23]. Although the antibiofilm effects of many natural compounds on specific bacterial strains have been proven [24,25], the underlying molecular mode of action has not been well explained. Several natural compounds, such as salicylic acid, cinnamic acid, and zosteric acid, have been shown to have antibiofilm activity at sub-lethal concentrations by reducing bacterial adhesion, affecting biofilm structural development and promoting biofilm detachment [26,27,28,29,30]. In one study, the antibiofilm compounds triggered the accumulation of reactive oxygen species (ROS) within the biofilm colonies [27,31,32]. ROS are recognized as intracellular messengers in cell growth and differentiation [33], as well as in biofilm formation [34]. All of these findings suggest that the antibiofilm compounds affect biofilm formation by fine-tuning the threshold level for oxidative stress. By combining a protein pull-down assay and mass spectrometry analyses, Cattò and co-workers [27,28] elucidated the interactions of salicylic and cinnamic acids with the protein WrbA. WrbA is a flavoprotein-bearing quinone oxidoreductase activity that is capable of reducing quinone substrates via a concerted two-electron mechanism [35,36].

Despite the interest in WrbA for its potential as a biofilm modulator and an antibiofilm compound biomarker, its role in biofilm development is still poorly investigated. Recently, Moshiri et al. [37] observed that WrbA inhibition resulted in a reduced capacity of *Salmonella enterica* serovar Typhi to efficiently adopt the biofilm lifestyle into microtiter systems. Although microtiter well plates are useful tools for studying the onset of the adhesion phase in biofilm development, they are limited in mature biofilm formation [38]. Furthermore, the authors speculated that the oxidoreductase activity of WrbA could be responsible for its role in biofilm formation [37]. There are still many unanswered questions about the contributions of WrbA to a wide range of biological functions, especially in terms of biofilm maturation and oxidative stress.

In this study, *Escherichia coli* K-12 strains BW25113 and BW25113-∆*wrb*A were used as model systems to investigate the key roles of WrbA in biofilm maturation, biofilm imbalance between the oxidant burden and the antioxidant defenses, and biofilm susceptibility to chemical treatments. To focus only on the fast-growing and maturation phases of biofilms, the cell adhesion phase was bypassed by forcing the bacteria to attach to a polycarbonate surface using the membrane-supporting biofilm culturing system [39]. Functional validation was also performed for the WrbA activity against two well-known antibiofilm compounds, namely cinnamic and salicylic acids [27,28]. This validation proved that the interaction and modulation of the WrbA function controlled the antibiofilm properties of the compounds being tested, strengthening the key role of WrbA in biofilm formation.

## 2. Materials and Methods

### 2.1. Escherichia Coli Strain and Preparation of the wrbA-mutant

The *Escherichia coli* strain JW0989 (F-, Δ(*ara*D-*ara*B)567, Δ*lac*Z4787(::*rrn*B-3), λ-, Δ*wrb*A748::kan, *rph*-1, Δ(*rha*D-*rha*B)568, *hsd*R514) was used for the preparation of the *wrb*A mutant [40]. JW0989 is an *E. coli* K-12 BW25113-derivative strain that contains the *wrb*A gene (GenBank acc. n.: CP009273.1, region 1063164-1062568) replaced by a kanamycin resistance cassette. Luria–Bertani (LB) medium (L3022, Sigma-Aldrich, St. Louis, MO, USA) was used for the *wrb*A mutant preparation. An electrocompetent aliquot of JW0989 was electroporated with the pCP20 helper plasmid [41], which has a temperature-sensitive origin of replication, confers ampicillin resistance, and encodes the FLP recombinase. The ampicillin-resistant transformants were selected at 30 °C on an agar plate. Thirty-two transformant colonies, as well as JW0989 and BW25113 as negative and positive controls, respectively, were replicated on agar plates with and without kanamycin. The plasmid pCP20 was cured by growing the kanamycin-sensitive colonies at 42 °C without antibiotics, starting each growth round from the previously grown round. A test for the ampicillin sensitivity confirmed that pCP20 was cured and the resulting *wrb*A mutant strain was named BW25113-∆*wrb*A. The strains were PCR-validated using the primers described in Table 1. The strains were stored at −80 °C in a 20% glycerol and 2% peptone solution and were routinely grown in tryptic soy broth (TSB, Sigma-Aldrich) at 37 °C for the subsequent experiments. Different transformant colonies were used independently for the biological replicates in all experiments.

### 2.2. E. coli Planktonic Growth

The planktonic growth rates of both BW25113 and BW25113-∆*wrb*A strains were monitored in 96-well microtiter plates. In total, 200 µL of TSB containing 10^6^ cells/mL was aliquoted in the microtiter wells and incubated at 37 °C for 24 h. Growth growth rates were followed by measuring the absorbance at 600 nm (A_600_) every 15 min using the Infinite 200 PRO Microplate Reader (Tecan). Absorbance-based growth kinetics were constructed by plotting the A_600_ of suspensions minus the A_600_ of the non-inoculated medium against the incubation time. The polynomial Gompertz model was used to fit the growth curves, while the duration of the lag phase (λ), the maximum specific growth rate (µ_m_), and the maximum growth (Y_M_) were calculated using GraphPad Prism software (version 5.0, San Diego, CA, USA). Three biological replicates were performed for each condition.

### 2.3. E. coli Biofilm Growth

Membrane-supported biofilms of both BW25113 and BW25113-∆*wrb*A strains (hereafter referred to as BW25113 and BW25113-∆*wrb*A biofilms) were prepared following the method reported by Anderl et al. [44]. Briefly, 50 µL of 10^7^ cells/mL bacterial suspensions in the exponential phase was used to inoculate sterile black polycarbonate filter membranes (0.22 mm pore size and 25 mm in diameter, Whatman) resting on tryptic soy agar (TSA, Sigma Aldrich) plates. Membranes were then incubated at 37 °C for 72 h. The media were replaced every 24 h to avoid any change in the experimental conditions that could affect the biofilm growth (e.g., nutrient deficiency and metabolic product build-up).

At 6, 12, 18, 24, 36, 48, 60, and 72 h, two polycarbonate membranes were removed from Petri dishes and transferred to 15 mL PBS. Sessile cells were removed from the membrane surface via 30 s vortex mixing and a 2 min sonication bath (Branson 3510, Branson Ultrasonic Corporation, Dunburry, CT, USA), followed by an additional 30 s vortex mixing. To remove bacterial aggregates, cell suspensions were homogenized by applying two cycles at 14,500× *g* rpm for 30 s (T 10 basic Ultra-Turrax), followed by 30 s vortex mixing. Ten-fold serial dilutions of the resulting cell suspensions were plated on TSA and incubated overnight at 37 °C. Counts of colony-forming units (CFUs) were determined as reported by Miller [45]. Data are reported as CFU/cm^2^ and normalized as the CFU inoculated on the membrane at 0 h. Three biological replicates were performed for each condition.

Biofilms collected at 48 and 60 h were carefully covered with a layer of Killik cryostat embedding medium (Bio-Optica, Milan, Italy) and placed at −80 °C until completely frozen. Frozen samples were sectioned at −19 °C using a Leica CM1850 cryostat, then the 10-μm-thick cryosections were mounted on Superfrost Plus microscope slides (ThermoFisher Scientific). The cryosections of the biofilms were stained with the lectin Concanavalin A-Texas Red conjugate (ConA, Invitrogen, Italy) to visualize the polysaccharide component of the extracellular polymeric matrix (EPM). The cryosections were observed in epifluorescence mode using a Leica DM 4000 B microscope with a 20X or 40X dry objective. The biofilm thickness was measured for each image at three different locations, which were randomly selected along the profile. These measurements were used to calculate the average thickness and the associated standard deviation. ImageJ software [46] was used to overlay the images acquired in bright field (the whole biofilms) and epifluorescence (the EPM) modes and to measure the biofilm thickness.

The biofilms were also grown on TSA supplemented with 25 μg/mL congo red (CR) to detect amyloid curli in both BW25113 and BW25113-∆*wrb*A strains. Biofilms grown on CR agar medium were collected at 48 and 60 h. The biofilm patina was collected with a sterile loop and placed on a microscope slide. Images were taken in both epifluorescent and bright field modes as previously described for the cryosections.

### 2.4. Level of Oxidative Stress within the Biofilm

BW25113 and BW25113-∆*wrb*A biofilms were grown as previously described in Section 2.3. Biofilm samples were collected at 48 and 60 h, corresponding to the fast-growing phase and the maturation phase, respectively. The biofilms were dislodged from the membranes and the biomass suspension was transferred into screw cap vials containing approximately 100 μL of glass beads with different sizes (50% beads with diameters ranging from 425 to 600 µm and 50% beads with diameters < 106 µm). Cells were mechanically disrupted using a Precellys bead beater (Bertin Instrument, France), whereby six cycles of 30 s at 6500× *g* rpm were performed, with 30 s cooling periods between cycles. Each sample was then centrifuged (11,000× *g*) for 30 min. The supernatants were recovered, and the level of reactive oxygen species (ROS) was measured using the 2′,7′-dichlorodihydrofluorescein diacetate (H_2_DCFDA) assay according to Jakubowski et al. [47]. Briefly, 5 mM H_2_DCFDA was added to the sample to a final concentration of 10 mM and samples were incubated at 30 °C. After 30 min, the fluorescence was measured using the Infinite 200 PRO Microplate Reader (Tecan) with excitation and emission at 488 and 520 nm, respectively. Fluorescence values were normalized to the number of cells and the means were reported. Three biological replicates were performed for each condition.

The levels of oxidative stress within the biofilms were also visualized through the combination of biofilm sectioning and imaging by fluorescent microscopy. Cryosections of 48- and 60-h-old biofilms were obtained as described in Section 2.3. The biofilm cryosections were stained with 5 μM CellROX Green dye (ThermoFisher Scientific, Italy) for 30 min at room temperature. Samples were observed using a Leica DM 4000 B microscope with a 20X dry objective. The sections were viewed both in bright field and in epifluorescence mode. ImageJ software [46] was used to overlay the brightfield images with the fluorescence images.

### 2.5. Biofilm Catalase Activities

The catalase activities were quantified using the protocol previously described by Iwase et al. [48]. Briefly, biofilms samples were collected at 48 and 60 h, and cells suspensions in PBS were obtained as previously described in Section 2.3. The cell suspensions were diluted to a final concentration of 10^6^ cell/mL. In total, 100 μL of the cell suspensions was mixed with 100 μL of 1% Triton X-100 and 100 μL of undiluted hydrogen peroxide (30%) in a Pyrex tube (13 mm diameter × 100 mm height). After 10 min of incubation at room temperature, the height of the O_2_-forming foam in the test tube was measured using a ruler.

In order to distinguish between the activities of the heat-labile catalase HPI and heat-stable catalase HPII, the bacterial suspension was divided into 2 identical aliquots. One aliquot of the bacterial suspension was subjected to heat treatment at 55 °C for 15 min. Then, the residual catalase activity following heat treatment was subtracted from the total catalase activity to determine the activity of the heat-labile catalase HPI.

To quantify the catalase activity, a calibration curve was plotted with the defined unit of catalase activity. The experiments were conducted in triplicate.

### 2.6. Biofilm Susceptibility Assay

A volume containing 2-chlorobenzoic acid and H_2_O_2_ stock solutions was added to molten culture medium TSA to create a biocide-amended agar for biofilm experiments. The final biocide concentration in biofilm assays was 6 mM 2-chlorobenzoic acid and 4.5 mM H_2_O_2_ [49]. Biofilms of both strains, which were 48 and 60 h old, were aseptically transferred to either biocide-containing agar, biocide and 10 mM glutathione-containing agar, or a control plate, then incubated overnight at room temperature. After this time, the biofilm was collected, physically disaggregated, 10-fold serially diluted, and plated on TSA as reported above. The antimicrobial efficacy is expressed as the log_10_ reduction, which represents the killing of biofilm-dwelling cells after the treatment. The log_10_ reduction was calculated relative to the cell count in the control samples without biocides. All antimicrobial experiments were conducted in duplicate.

### 2.7. E. coli Crude Extracts

*E. coli* cells were collected from −80 °C glycerol stock and grown in 250 mL LB medium for 16 h at 37 °C under 150 rpm orbital agitation. The culture was divided in 3 aliquots and cells were collected via 5000× *g* centrifugation at 4 °C for 15 min. Each aliquot of collected cells was washed by suspension in 6 mL of phosphate-buffered saline (PBS; 10 mM phosphate, 2.7 mM potassium chloride, 137 mM sodium chloride, pH 7.4) and centrifuged under the same conditions. The cell pellet was washed another time in the same manner as before, then was well drained, weighted (~0.28 g), and stored at −80 °C. For the enzyme assay, the cell pellet was thawed on ice and suspended in 3 volumes of 50 mM Tris-HCl at pH 7.2. The suspended cells were broken by sonication in a Soniprep 150 instrument (MSE, London, UK) equipped with a small tube probe, involving five 30 sec sonication cycles with an amplitude of 18 μm followed by 1 min on-ice cooling periods. The cell lysate was centrifuged for 30 min at 16,000× *g* at 4 °C and the supernatant (i.e., crude extract) was collected and stored in aliquots at −20 °C. The amount of protein was quantified in crude extracts by Bradford assay [50] using bovine serum albumin as the standard.

### 2.8. NADH-Dependent Oxidoreductase Activities

The oxidoreductase activity of the *E. coli* crude extracts was detected using reduced β-nicotinamide adenine dinucleotide (NADH, N8129 Sigma Aldrich) as the electron donor substrate and *p*-benzoquinone (*p*-BQ; 12,309 Sigma Aldrich) as the electron acceptor substrate (NADH:*p*-BQ oxidoreductase activity), or the 2,6-dichlorophenolindophenol (DCPIP; D1878 Sigma Aldrich) redox dye as an electron acceptor (NADH:DCPIP oxidoreductase activity) using the methods described by Patridge and Ferry [36], with some modifications. The measurements for the NADH:*p*-BQ oxidoreductase activity were carried out in 1 mL assay mixtures composed of 50 mM Tris-HCl (pH 7.2), 0.3 mM *p*-BQ (from 3 mM stock in water), 0.2 mM NADH (from 4 mM stock in 50 mM Tris-HCl, 1 mM EDTA, pH 7.7), and 10 µL (protein amount, 5–20 µg) of crude extract sample, which was properly pre-diluted with the assay buffer.

In the case of NADH:DCPIP oxidoreductase activity measurements, 0.1 mM DCPIP (from 2 mM stock in assay buffer) replaced *p*-BQ. The enzyme reaction was started by the addition of the crude extract sample and was monitored by measuring the NADH consumption (λ = 340 nm; ε = 6.22 mM^−1^ cm^−1^) or the formation of reduced DCPIP (λ = 610 nm; ε = 18.1 mM^−1^ cm^−1^) using a spectrophotometer (Lambda 2, Perkin Elmer, USA) equipped with a Peltier system (PTP-6, Perkin Elmer, USA) to maintain the temperature at 25 °C. To obtain the enzyme reaction rate, the rate for the first 10 sec recorded after the addition of the last component was subtracted from the non-enzymatic rate recorded before the addition of the last component. In this work, the enzyme unit (U) of NADH:*p*-BQ oxidoreductase (or NADH:DCPIP oxidoreductase activity) is the amount of enzyme function that consumes 1 μmol of NADH (or produces 1 µmol of DCPIP) in a minute of reaction at 25 °C (pH 7.2). Enzyme activity data were expressed as the specific activity normalized to the amount of protein detected by Bradford assay (U/mg). The percentage of WrbA contribution (activity_WrbA_) to the measured activity was calculated as follows:$${\mathrm{activity}}_{\mathrm{WrbA}}=\frac{\left({\mathrm{activity}}_{\mathrm{BW}25113}-{\mathrm{activity}}_{\mathrm{BW}25113-\Delta wrbA}\right)}{{\mathrm{activity}}_{\mathrm{BW}25113}}\times 100$$
where activity_BW25113_ and activity_BW25113-∆*wrb*A_ are the enzyme activities (U/mg) measured in crude extracts prepared from BW25113 and BW25113-∆*wrb*A, respectively.

### 2.9. Purification of WrbA

The *E. coli* strain JW0989 ASKA (-) harboring the expression plasmid pCA24N*wrb*A(-) was used to overexpress the WrbA polyHis tagged to its N-terminus (WrbA_JW_), which belongs to an *E. coli* strain collection suitable for protein functional studies [51] and was supplied by the National BioResource Project (NBRP-*E. coli*) at the Microbial Genetics Laboratory of the National Institute of Genetics (Yata, Mishima, Shizuoka; Japan). The sequence of overexpressed WrbA_JW_ is identical to accession number P0A8G6, with MRGSHHHHHHTDPALRA replacing the start methionine residue and a GLCGR extension at its C-terminus. The cells were cultured to A_600_ of 0.8 at 37 °C in an orbital shaker (150 rpm), then cooled and induced using 1 mM IPTG during a 20 h incubation at 25 °C. Cells were harvested by centrifugation (4000× *g*, 15 min, 4 °C), washed 2-fold with degassed PBS, and stored at −80 °C until crude extract preparation. Collected cells were suspended in 3 volumes of degassed extraction buffer (50 mM NaH_2_PO_4_, 100 mM NaCl, 1 mM EDTA, pH 7.2) containing 0.02 mg/mL lysozyme, incubated for 30 min on ice and disrupted by sonication in a Soniprep 150 instrument (MSE, London, UK) equipped with a 9.5 mm probe, involving seven 30 sec sonication cycles with an amplitude of 16 μm, followed by 2 min on-ice cooling periods. Insoluble debris material was removed using two consecutive 30 min centrifugation periods (16,000× *g* at 4 °C) and the supernatant (i.e., crude extract) was submitted to ion metal affinity chromatography (IMAC) in an HPLC system equipped with a 5 mL column (Ni-NTA Superflow^®^, H-PR cartridge, IBA GmbH, Goettingen, Germany) and a dual-wavelength absorbance detector (2487; Waters Corporation, Milford, MA, USA.) for inline monitoring at 280 and 450 nm. The IMAC was carried out in degassed 50 mM NaH_2_PO_4_, 100 mM NaCl, pH 7.2 at 2 mL/min flow, and a biphasic gradient (0.18 M imidazole in 70 mL, then 0.40 M imidazole in 30 mL) was applied to elute WrbA_JW_ at 0.25 M imidazole. The eluted fraction was incubated for 15 min on ice in the presence of 5 mM FMN and desalted against 50 mM NaH_2_PO_4_ and 100 mM NaCl (pH 7.2) using multiple steps in an ultrafiltration centrifugal 15 mL filter (4000× *g* at 4 °C, cutoff 10000; Amicon^®^, Merck Millipore, Burlington, MA, USA.). Desalted eluted WrbA_JW_ was frozen in liquid nitrogen and stored in aliquots at −80 °C. The quality of overexpression and purification was checked using SDS-PAGE analyses, and the WrbA_JW_ functionality (1032 ± 165 U/mg) was assessed using determinations of NADH:*p*-BQ oxidoreductase activity.

To evaluate the effects of antibiofilm molecules on the NADH:*p*-BQ oxidoreductase activity of WrbA_JW_, 4 mM cinnamic acid or salicylic acid (from 40 mM stocks in assay buffer, finally adjusted to pH 7.2) was added to the assay mixture (see Section 2.8). The order of addition was: assay buffer, 20 µL WrbA_JW_ (~0.1 µg), tested molecule, NADH, and *p*-BQ. In control assay reactions, the tested molecule was omitted.

### 2.10. Statistical Analysis

ANOVA or t-test analysis using XLSTAT software (Version 2021.1, Addinsoft, France) was applied to statistically evaluate any significant differences among the samples. The ANOVA analysis was carried out after verifying the data independence (Pearson’s Chi-square test), normal distribution (D’Agostino–Pearson normality test), and homogeneity of variances (Bartlett’s test). Tukey’s honestly significant different test (HSD) was used for pairwise comparison to determine the significance of the data. Statistically significant results are depicted by *p*-values < 0.05.

## 3. Results and Discussion

### 3.1. WrbA Is Not Essential for the Planktonic Growth but It Affects the Biofilm Growth

The use of a mutant strain lacking the *wrb*A gene is a powerful tool for advancing the understanding of the roles of WrbA in *E. coli* biofilm physiology and behavior. Due to its sequenced genome and growth plasticity, *E. coli* has become one of the most important model organism workhorses in biology and biotechnology [52,53]. As a model organism, *E. coli* has been widely used in studies of molecular biology, physiology, genetics, and evolution, as well as pharmaceutical, genetic engineering, and biotechnology studies [54].

In order to investigate the role of WrbA in biofilm formation, it must be verified that the lack of WrbA does not affect the essential cellular functions such as the cellular growth in the planktonic form. The *E. coli* strains BW25113 and BW25113-∆*wrb*A had similar planktonic growth curves with no statistical differences in the lag-phase durations (BW25113 2.75 ± 0.27 h; BW25113-∆*wrb*A 2.13 ± 0.48 h), maximum growth rates (BW25113 0.10 ± 0.03 A_600_/h; BW25113-∆*wrb*A 0.09 ± 0.01 A_600_/h), and maximum growth values (BW25113 0.99 ± 0.03 A_600_; BW25113-∆*wrb*A 1.07 ± 0.01 A_600_) (Appendix A). The observation that the lack of WrbA did not impair the planktonic growth suggested that any phenotypical changes to the BW25113-∆*wrb*A biofilm could be associated with the *wrb*A deletion. By performing the complementation on the mutant strain, it can be ensured that the observed mutant phenotype is actually from the loss of *wrb*A and not from secondary mutations that may have occurred during the creation of the mutant strain. A genetic complementation strategy was not undertaken because: (i) the single-gene deletion was kept in the parental strain and no transduction steps were adopted; (ii) the genetic complementation requires additional plasmid sequences and the subsequent use of plasmid-related antibiotics (and the possible promoter inducer), which introduces new condition in the phenotype analysis; (iii) the chemical complementation was successful (see Section 3.3).

Figure 1 displays the biofilm growth curves for each *E. coli* strain. Overall, the numbers of biofilm-dwelling increased until the 60th hour of incubation and then plateaued afterwards. The results indicated that the mutant strain BW25113-∆*wrb*A produced more biofilm-dwelling cells during the first 48 h than the wild-type strain BW25113. However, after 60 h, the number of BW25113-∆*wrb*A biofilm-dwelling cells was lower than that of the wild-type by two orders of magnitude. Interestingly, the highest increases in CFU/cm^2^ for both the wild-type (+4.9 CFU/cm^2^) and the mutant (+1.5 CFU/cm^2^) strains happened between 48 and 60 h. The 48 and 60 h time points displayed the transition between the exponential phase and the stationary phase, corresponding to growing and mature biofilms, respectively. Significant physiological, morphological, and gene expression changes occurred when growing cells entered a plateau, indicating the transition to a mature biofilm [55,56]. Thus, the effect of WrbA was investigated during this critical transitional time. The 48- and 60-h-old biofilms were taken as being representative of the fast-growing and maturation phases, respectively, and were used for further investigations.

In line with the biofilm growth results, biofilms cryosections combined with microscopy observations revealed that BW25113-∆*wrb*A 60-h-old biofilms (thickness of 134 ± 22 µm) were significantly thinner than those of BW25113 (thickness of 262 ± 32 µm), as shown in Figure 2. The yield reduction in biofilm thickness was 48.6%. Although both wild-type and mutant biofilms retained similar morphological patterns, the red signal level corresponding to the EPM polysaccharide fraction was less in the BW25113-∆*wrb*A biofilms than in the BW25113 biofilms in both the fast-growing and maturation phases. Altogether, the results indicated that the lack of WrbA reduced not only the number of biofilm-dwelling cells, but also the biofilm thickness and the EPM polysaccharide content. The EPM is one of the key elements in the establishment and maintenance of the biofilm structure and properties. Several exopolysaccharides were identified in the *E. coli* EPM as key components of the biofilm matrix, including cellulose, poly-β-1,6-N-acetyl-D-glucosamine, and colanic acid [55]. In *E. coli* K12, colanic acid did not contribute to surface adherence, but rather to the development of robust biofilms and the maintenance of the biofilm’s 3D architecture [57]. It is well known that the interactions between EPM and ROS alleviate the oxidative stress experienced by a biofilm [58]. In fact, *E. coli* cells deficient in colonic acid production were more susceptible to H_2_O_2_ than their wild-type counterparts, indicating that the polysaccharide played a role in protecting the bacterium from oxidative stress [59]. An increase in ROS production within the biofilm might decrease the EPM and consequently the biofilm formation, as has been demonstrated for *Staphylococcus aureus* [60].

The amyloid curli fimbriae have been identified as important EPM components of *E. coli* biofilms [55]. For this reason, CR was used to qualitatively assess the effects of WrbA on amyloid protein production in both BW25113 and BW25113-∆*wrb*A biofilms (Figure 3). Although cellulose is another important component of *E. coli* EPM that can be stained with CR, *E. coli* BW25113 does not produce cellulose due to a point mutation in the cellulose biosynthesis protein BcsQ [61]. Therefore, the CR fluorescent signals visualize only the amyloid proteins. BW25113 EPM showed greater amyloid signals than BW25113-∆*wrb*A biofilms, suggesting an effect of WrbA on curli fiber production. Curli production is highly responsive to stressful conditions, and it depends on the CsgD transcription activator [62]. Soo and Wood [63] observed significant reductions in curli production and *csg*D transcript under oxidative stress.

Interestingly, the two *E. coli* strains differed in biofilm morphologies at 60 h. The BW25113 colony biofilm displayed a central core surrounded by a prominent peripheral area with irregular outer boundaries (Figure 3). On the contrary, in the BW25113-∆*wrb*A biofilm, the central core was surrounded by a peripheral area with a uniform boundary profile (Figure 3). The mechanomorphogenesis of biofilms is strictly connected to the environmental conditions (e.g., the characteristics of the substrate), the stress dynamics during biofilm development, and the biofilm’s physiological heterogeneity [64]. Phenotypic variants of biofilms may mirror differences in the biofilm physicochemical characteristics [65]. For example, in *Vibrio cholera*, wrinkled morphologies are associated with the mechanical instability [66]. Fei and collaborators demonstrated that stress insurgence caused by nutrient depletion created a non-uniform growth and subsequent anisotropic compressive stress in the outer region of the biofilm [67].

### 3.2. WrbA Affects the Biofilm Oxidative Stress Profile and Alternative Antioxidant Defenses

The BW25113-∆*wrb*A biofilm growth curves suggested a sensitive step—corresponding to the transition from the fast-growing phase at 48 h to the maturation phase at 60 h—with a significant slowdown in biofilm development. Since we believed that WrbA was potentially contributing to the cells’ oxidative stress resistance [36,68,69], we investigated whether the sensitive growth step was related to the shift in the oxidative stress level in the biofilms. At 48 h, the level of oxidative stress in the BW25113-∆*wrb*A biofilms was higher than that of the BW25113 biofilms, as shown in Figure 4A. In contrast, in the mature biofilms at 60 h, BW25113-∆*wrb*A biofilms appeared to be under less oxidative stress than the wild-type biofilms (Figure 4B). To study the spatial distribution of ROS in the unperturbed biofilm communities, cryosections combined with CellROX probe staining were performed on growing (48 h) and mature (60 h) biofilms (Figure 4C). The CellROX probe allows the in-situ localization of the oxidative stress level in biofilm cross-sections by turning fluorescent as it binds to the ROS-oxidized DNA. The growing BW25113 and BW25113-∆*wrb*A biofilms showed high levels of oxidative stress throughout the biofilm community, with peaks at the air-exposed surfaces. In contrast, the mature biofilms of the mutant strain showed fewer green signals than those of the wild type. In BW25113-∆*wrb*A biofilms, the green signal was localized in the air-exposed surface of the colony. This result was consistent with the highest oxygen levels observed at the upper part of the biofilm [70,71]. Oxidative stress could also have been created by altering the electron transport or by increasing the production of redox-active metabolites [72].

Overall, these results suggested that the imbalance between the oxidant burden and antioxidant defenses in BW25113-∆*wrb*A biofilm during the fast-growing phase could provide the selective pressure to increase the biofilm-forming capacity of the mutant strain. Thus, during the fast-growing phase, the chronic endogenous oxidative events in BW25113-∆*wrb*A biofilms generated a stress condition that invoked the bacterium to grow the biofilm better than its wild-type counterpart. As the biofilm reached the maturation phase, a reduced metabolic activity and enhanced redox buffering properties may have compensated for the lack of WrbA, explaining the low amount of BW25113-∆*wrb*A biofilm-dwelling cells. Several studies have shown the significant correlation between the oxidative stress and the biofilm formation [73,74,75]. In fact, reactive oxygen species (ROS) influence not only the biofilm physiology, but also the biofilm properties, structure, and morphology [76]. As a result of certain mechanisms that are not well understood yet, ROS can promote or inhibit biofilm formation. In the case of *Pseudomonas aeruginosa* and *E. coli* treated with ROS-generating aminoglycoside antibiotics [77] and *Campylobacter jejuni* grown under aerobic conditions [78], ROS promote biofilm formation. In a previous study, chronic sub-lethal oxidative events promoted the sessile growth of the soil model bacterium *Azotobacter vinelandii* [49]. Meanwhile, in other studies, ROS inhibited biofilm formation, for example in the case of *Listeria monocytogenes* [79] and *E. coli* K-12 (ATCC25404) [27]. One of the main clues that connects biofilm formation to oxidative stress is the involvement of the same regulators in both oxidative stress response and biofilm development. For example, the protein OxyR that is involved in the oxidative stress response is also involved in biofilm formation, as demonstrated by several studies that used mutants for *oxy*R [80,81,82]. Sigma factor RpoS (σ^s^ or σ^38^), a stress response activated by ROS concentration, controls several genes involved in biofilm formation [83] and is responsible for the upregulation of CsgD, which elicits the synthesis of curli and cellulose [37]. As confirmed by global expression studies, RpoS also controls *wrb*A expression under oxidative stress, acidity, salt stress, diauxic growth [36], and starvation [37]. It has been hypothesized that the inhibition of WrbA may lead to ROS accumulation, interfering with the quorum-sensing system and biofilm formation [27].

Regardless of the source of the oxidants, our results suggested that the lack of WrbA activated alternative and efficient antioxidant defenses, which drastically reduced the oxidative stress level within the biofilms. To prove this, the catalase activities of BW25113 and BW25113-∆*wrb*A biofilms were measured at 48 and 60 h (Figure 5). *E. coli* has two catalases, HPI and HPII. The HPI is induced during logarithmic growth in response to a low concentration of hydrogen peroxide and depends on OxyR. In contrast, HPII is not peroxide-inducible and is regulated by the global stress regulator RpoS [84,85]. The *rpo*S gene is known as the alternative sigma (σ) factor, which controls the expression of a large number of genes involved in various stress responses, nutrient scavenging, expression of virulence factors, acid resistance, osmotic stress resistance, and synthesis of the cell’s structural components [86,87]. Several studies have focused on these two catalases for their pivotal roles in protecting cells against oxidative stress [88,89]. The results indicated that the activity of HPII is higher in mature BW25113-∆*wrb*A biofilms than in mature BW25113 biofilms. As the expression of catalase HPII encoded by *katE* is highly RpoS-dependent, catalase activity could also be used to assess RpoS activity [48,90], whereby higher catalase activity means higher RpoS activity. Corona-Izquierdo and Membrillo-Hernández [91] reported that *E. coli rpo*S mutant strains exhibited increased production of biofilm, especially in the fast-growing phase. The findings suggested the key role of RpoS in controlling the biofilm formation and in preventing the synthesis of an extracellular factor that promotes the formation of biofilm during the fast-growing phase. Sheldon et al. [92] reported an enhanced biofilm biovolume in the *E. coli* O157:H7 *rpo*S-deficient strain compared to the wild-type counterpart. Liu and coworkers [93] proved that *rpo*S mutations in the endophytic bacterium *Serratia plymuthica* G3 resulted in significant enhancements in swimming motility, biofilm formation, and in production of the plant auxin indole-3–acetic acid. Recently, Zlatkov and Uhlin [94] demonstrated that the loss of RpoS function in extraintestinal pathogenic *E. coli* enhances colonization, citrate utilization, and citrate-complexed iron transport. All of these results suggested a negative control of RpoS on biofilm formation. Thus, the lack of WrbA in *E. coli* mature biofilms seemed to promote the activity of the RpoS-dependent HPII, which ultimately reduces ROS concentration. Overall, these findings suggested that the lack of certain defenses against oxidative stress in the mutant strain promotes compensatory measures to scavenge the excess of ROS in mature BW25113-∆*wrb*A biofilms. Future research will be devoted to studying the diverse, temporally structured gene expression responses triggered by the lack of *wrb*A in order to fully explain the phenotype of the mutant strain.

### 3.3. WrbA Affects the Biofilm Tolerance to H_2_O_2_ but Not to 2-Chlorobenzoic Acid

To study the substantial contribution of WrbA to defending *E. coli* against ROS, both BW25113 and BW25113-∆*wrb*A mature biofilms were challenged with H_2_O_2_ (Figure 6). Overall, the BW25113-∆*wrb*A biofilms were more susceptible to H_2_O_2_ than BW25113 biofilms. However, the tolerance of BW25113-∆*wrb*A biofilms to H_2_O_2_ increased with the biofilm as it reached maturity. This phenomenon was expected, considering the barrier created by the extracellular polymeric substances, the physiological state of biofilm-dwelling cells, and the sub-population of resistant phenotypes that characterize mature biofilms [95,96,97].

In addition to H_2_O_2_, the wild-type and mutant biofilms were challenged with 2-chlorobenzoic acid, an antimicrobial compound that does not generate ROS (Figure 6). Several studies have reported that the toxicity of phenols (including chlorinated phenols) arises from their lipophilic character, which favors the interaction with cell membranes [98,99]. Both BW25113 and BW25113-∆*wrb*A showed the same tolerance to 2-chlorobenzoic acid. Thus, the difference in the biocidal mode of action between H_2_O_2_ and 2-chlorobenzoic acid could explain the difference in the susceptibility of BW25113-∆*wrb*A biofilms.

We also investigated the possibility of adding glutathione to chemically complement the mutation of BW25113-∆*wrb*A, thereby restoring the antimicrobial tolerance of biofilms formed by the mutant strain (Figure 6). When glutathione was added during the antimicrobial challenge, the H_2_O_2_ tolerance of the BW25113-∆*wrb*A biofilms increased. Significant upregulation of WrbA was observed in *Salmonella enterica* serovar Enteritidis ATCC 4931 biofilms exposed to benzalkonium chloride [100] and in resistant *E. coli* cells challenged with sub-lethal concentrations of olaquindox [101]. The gene *wrb*A was upregulated in persister cells, suggesting the contribution of the WrbA in antimicrobial defense mechanisms [102,103,104]. Our results indicate that WrbA is instrumental in promoting biofilm tolerance to biocides. The increased tolerance is mainly due to the defensive role of WrbA against the oxidative stress caused by several antimicrobial compounds.

### 3.4. WrbA Is an Important Functional Component of the E. coli NADH-Dependent Oxidoreductase Activity That Is Inhibited by Cinnamic and Salicylic Acids

The effects of the absence of WrbA on biofilm features must be related to the molecular function of the flavoprotein. Based on the sequence similarity and experimental evidence of the purified protein, an NADH-dependent oxidoreductase activity (EC 1.6.5.2) was related to WrbA [36]. In order to frame the participation of WrbA in the overall NADH-dependent oxidoreductase activity borne by the *E. coli-*soluble proteome, this activity was measured in the crude extracts of BW25113-Δ*wrb*A and was compared to that of BW25113. The NADH-dependent oxidoreductase activity was measured in the presence of *p*-benzoquinone (*p*-BQ) as the acceptor substrate (Figure 7A) or in the presence of the redox dye DCPIP as a final electron acceptor (Figure 7B). The NADH-dependent oxidoreductase activity was clearly detectable in the crude extracts prepared from cultures of the wild-type *E. coli* strain (BW25113), either by using *p*-BQ (22.2 ± 2.5 U/mg; *p* < 0.05) or DCPIP (3.4 ± 1.0 U/mg; *p* < 0.05). The NADH:DCPIP oxidoreductase activity was 6.6-fold lower than the NADH:*p*-BQ oxidoreductase activity. The NADH-dependent oxidoreductase activity detected in the crude extracts of BW25113-∆*wrb*A was lower than that of the wild-type strain (*p* < 0.05). These findings suggested that part of the NADH-dependent oxidoreductase activity in *E. coli* would be related to the WrbA function.

The contribution of WrbA to the NADH-dependent oxidoreductase activity was estimated by comparing the measured activity in the crude extracts of the BW25113 and BW25113-∆*wrb*A strains. The extrapolated contribution of WrbA was higher than 50% of the NADH-dependent oxidoreductase activity (Figure 7C). WrbA contributes mostly to the NADH-dependent oxidoreductase activity when the electron acceptor is *p*-BQ instead of DCPIP.

The data showed that the WrbA-mediated activity accounted for ~60% of the activity measured in the crude extract of the wild-type strain. The NADH:*p*-BQ oxidoreductase activity measured in the wild-type strain was 22.2 ± 2.5 U/mg; therefore, the calculated WrbA-mediated activity was ~15.3 U/mg. In the study by Patridge and Ferry [36], the NADH–*p*-benzoquinone oxidoreductase activity measured in purified recombinant WrbA was 990 ± 30 U/mg (in this work, WrbA_JW_ was 1032 ± 165 U/mg), which allowed us to estimate the WrbA amount to be ~22 µg per milligram of the total BW25113-soluble proteins. A similar extrapolation can be attained using the NADH:DCPIP oxidoreductase activity detected in BW25113 (3.4 ± 1.0 U/mg, this work) by comparing it with the value detected in BW25113-∆*wrb*A (this work, 1.7 ± 0.5 U/mg) and with that of the purified WrbA (160 ± 3 U/mg; Patridge and Ferry [36]). The last extrapolation allowed us to conclude that the WrbA-mediated NADH:DCPIP oxidoreductase activity was 50% of the activity detected in the wild-type strain, and that WrbA was estimated to be ~11 µg in one milligram of the total BW25113-soluble proteins. Overall, the results indicated that WrbA is the major component of *E. coli*-soluble proteins that drive the NADH-dependent oxidoreductase activity using *p*-BQ or DCPIP as electron acceptors. Moreover, the above extrapolations suggested that WrbA is an important active protein component of the *E. coli* crude extract-soluble proteins (1.1–2.2%). With respect to the electron acceptors, the WrbA contribution to the NADH-dependent oxidoreductase activity of *E. coli* crude extracts with DCPIP was less than with *p*-BQ (50.0% vs. 69.6%). Besides the differences in detectability due to the different assay procedures, the above difference could be due to the lower specificity of the redox dye DCPIP (assuming it has a general final electron acceptor role) with respect to *p*-BQ (assuming it has an acceptor substrate role). The amount of inferred active WrbA protein must be interpreted by considering that *E. coli* cells were collected during the planktonic stationary growth phase, when the *wrb*A transcript expression is supposed to be upregulated [105].

WrbA was identified as the unique protein in the *E. coli*-soluble proteome that was pulled-down by affinity chromatography on a resin functionalized with the active structures of zosteric acid (i.e., *p*-amino-cinnamic and *p*-amino-salicylic acid structures [27,28]. In order to evaluate the effects of the antibiofilm molecules on the function of WrbA, the NADH:*p*-BQ oxidoreductase activity of the purified recombinant WrbA was measured and compared to the one measured in the presence of cinnamic acid or salicylic acid (4 mM). The NADH:*p*-BQ oxidoreductase activity of WrbA was lower in the presence of the tested antibiofilm molecules, while the inhibition levels were evaluated to be 42.6 ± 10.9% and 51.6 ± 15.7% for cinnamic acid and salicylic acid, respectively (Figure 8).

Here, the inhibition of the WrbA function (quinone oxidoreductase activity) by cinnamic and salicylic acids was demonstrated for the first time. This result strongly suggested that the interaction and modulation of the WrbA function are the key aspects in biofilm formation, explaining the antibiofilm performance of cinnamic and salicylic acids. It can be hypothesized that other compounds bearing the cinnamate or the salicylate moieties, which were previously claimed as antibiofilm molecules, can act by negatively modulating the function of WrbA. WrbA was identified among the molecular targets of the salicylidene acylhydrazides, a class of antivirulence compounds able to block the function of the type III secretion system in several Gram-negative pathogens [106]. Recently, Moshiri and colleagues [37] identified WrbA as the molecular target of the compound T315, which was responsible for inhibiting the early stage of Salmonella enterica biofilm. Overall, these results point to WrbA as a promising biomarker for screening novel antibiofilm agents.

## 4. Conclusions

In this study, we demonstrated that the lack of WrbA did not impair the essential cellular functions for planktonic growth, but rather affected the biofilm formation through an ROS-dependent mechanism. The loss of WrbA function resulted in an ROS-sensitive phenotype, which is shown by the reductions of biofilm-dwelling cells, biofilm thickness, EPM polysaccharide content, and H_2_O_2_ tolerance once the maturation stage is reached. Endogenous oxidative events in the mutant strain BW25113-∆*wrb*A generated a stress condition to which the bacterium responded by increasing the catalase activity to compensate for the lack of WrbA. The subsequent reintroduction of *wrb*A in the mutant strain should restore the wild-type phenotype, proving that the observed effects in BW25113-∆*wrb*A biofilms are indeed caused by the loss of the gene of interest.

To further investigate the role of WrbA in biofilm formation, the oxidoreductase function of WrbA was measured in the presence of two well-known antibiofilm compounds, cinnamic and salicylic acids. Since the antibiofilm molecules inhibited the WrbA’s oxidoreductase function, the enzyme was functionally validated as a molecular target of salicylic and cinnamic acids, corroborating its role as a key player in biofilm formation. Prior to this work, the conclusive role of WrbA in *E. coli* biofilm development and oxidative stress had never been demonstrated. As WrbA homologues are present in the genomes of many pathogenic microorganisms, investigating the function and mechanism of this protein in oxidative stress resistance and biofilm formation will further the understanding of microbial infections and pathogenesis. Finally, the functional validation of WrbA can demonstrate its potential as a protein biomarker for screening novel antibiofilm compounds.

## Figures and Tables

**Figure 1 antioxidants-10-00919-f001:**
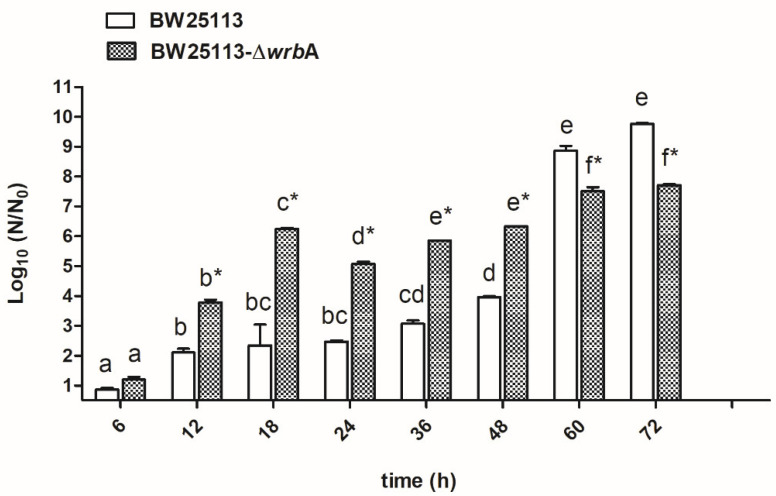
BW25113 and BW25113-∆*wrb*A biofilm growth curves. Data are expressed as log_10_ (N/N_0_), where N_0_ is the CFU/cm^2^ at the earliest time point and N is the CFU/cm^2^ at time t. Data represent the mean ± standard deviation of three independent measurements. The letters above each column indicate statistically significant differences (Tukey’s HSD, *p* ≤ 0.05) at the corresponding time steps, whereas the asterisks indicate a significant difference between the two strains at the corresponding time step (t-test *p* ≤ 0.05).

**Figure 2 antioxidants-10-00919-f002:**
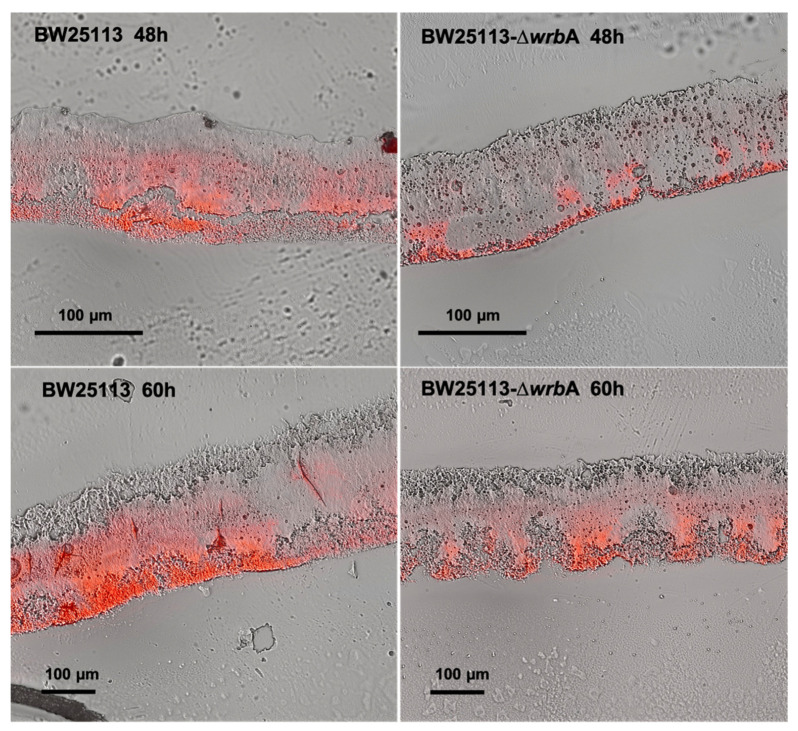
Images of cryosectioned BW25113 (first column) and BW25113-∆*wrb*A (second column) biofilms collected at 48 and 60 h. The images were obtained by overlapping the view acquired in bright field (the whole biofilms) and epifluorescence (the EPM stained with Texas Red-labeled Concanavalin A) modes.

**Figure 3 antioxidants-10-00919-f003:**
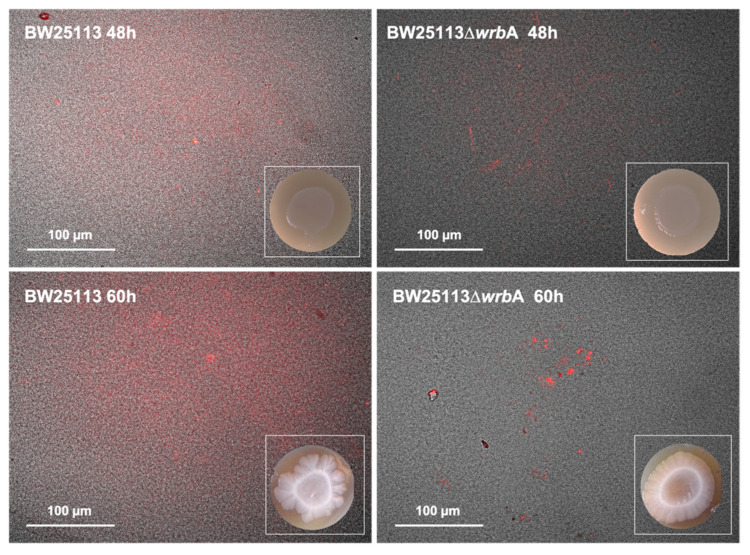
Biofilm morphology and visualization of extracellular curli in BW25113 and BW25113-∆*wrb*A biofilms after 48 and 60 h. Pictured were obtained by merging bright field and fluorescence images of biofilms grown on CR agar medium. Representative images are shown.

**Figure 4 antioxidants-10-00919-f004:**
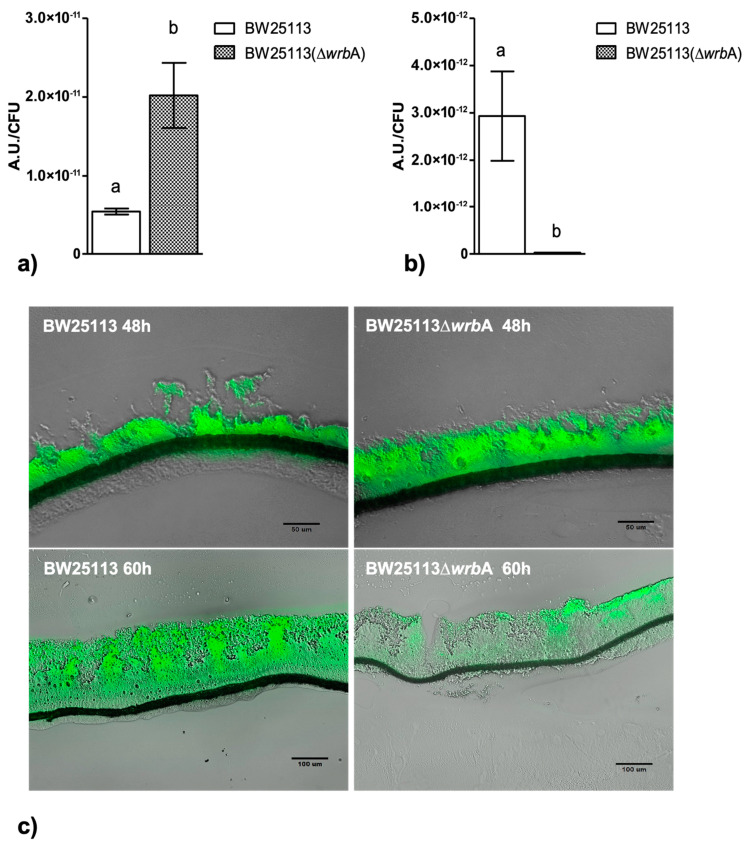
BW25113 and BW25113-∆*wrb*A oxidative stresses within the biofilm at 48 (panel **A**) and 60 h (panel **B**). Data are shown with the mean of the relative fluorescence (arbitrary unit, A.U.) normalized by the number of biofilm-dwelling cells with the standard deviation. The letters above each column indicate statistically significant differences (Tukey’s HSD, *p* ≤ 0.05) at the corre-sponding time steps. Panel **C** shows merged bright field and fluorescence images of cryosections representative of both BW25113 and BW25113-∆*wrb*A biofilms at 48 and 60 h. Samples were stained with the ROS-sensitive probe CellROX green.

**Figure 5 antioxidants-10-00919-f005:**
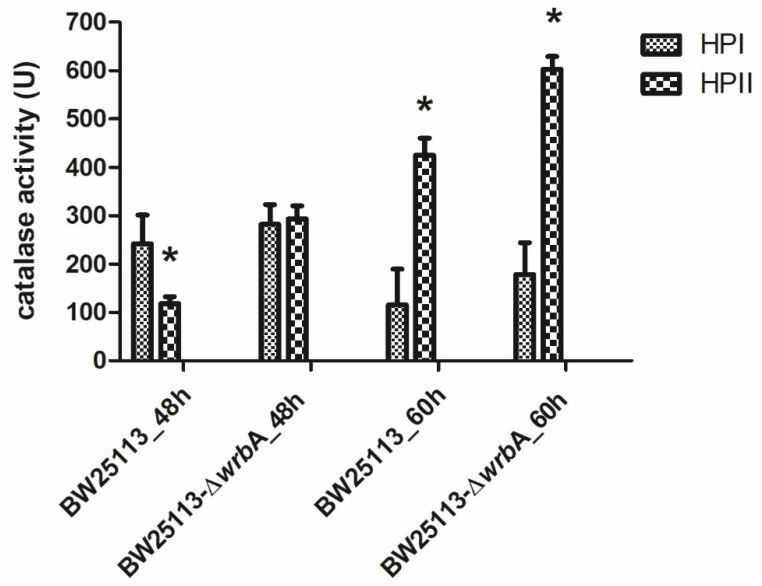
Catalase activities of both BW25113 and BW25113-∆*wrb*A biofilms at 48 and 60 h, shown as the mean values of catalase activity (unit) with the standard deviation of three independent measurements. The asterisks indicate statistically significant differences (t-test, *p* ≤ 0.05) between the two strains.

**Figure 6 antioxidants-10-00919-f006:**
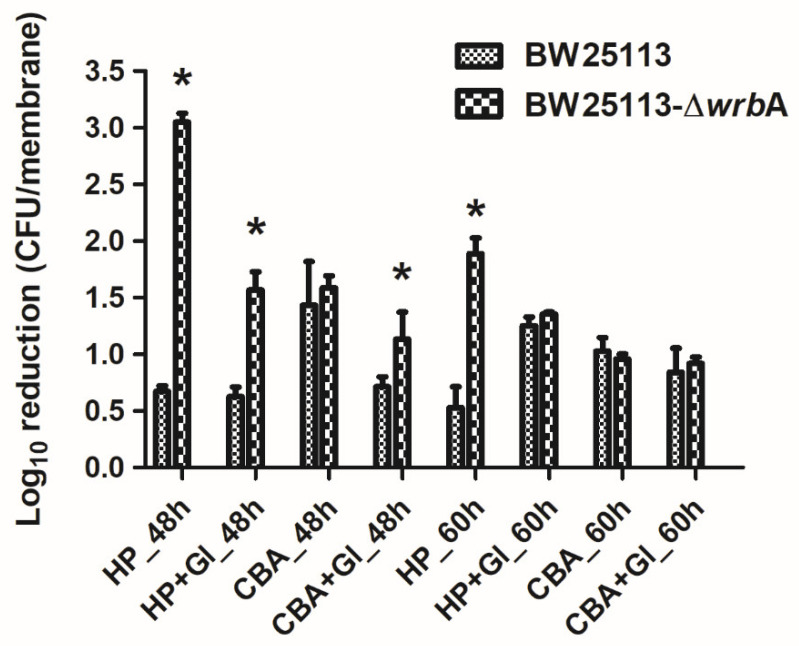
Susceptibility of BW25113 and BW25113-∆*wrb*A biofilms to antimicrobial agents, observed as a log_10_ reduction in the number of CFUs after exposure to 6 mM 2-chlorobenzoic acid (2-CBA) and 4.5 mM H_2_O_2_ (HP) with and without glutathione (Gl). Data are shown as the means ± standard deviation from three independent measurements. The asterisks indicate statistically significant differences (t-test, *p* ≤ 0.05) between the two strains.

**Figure 7 antioxidants-10-00919-f007:**
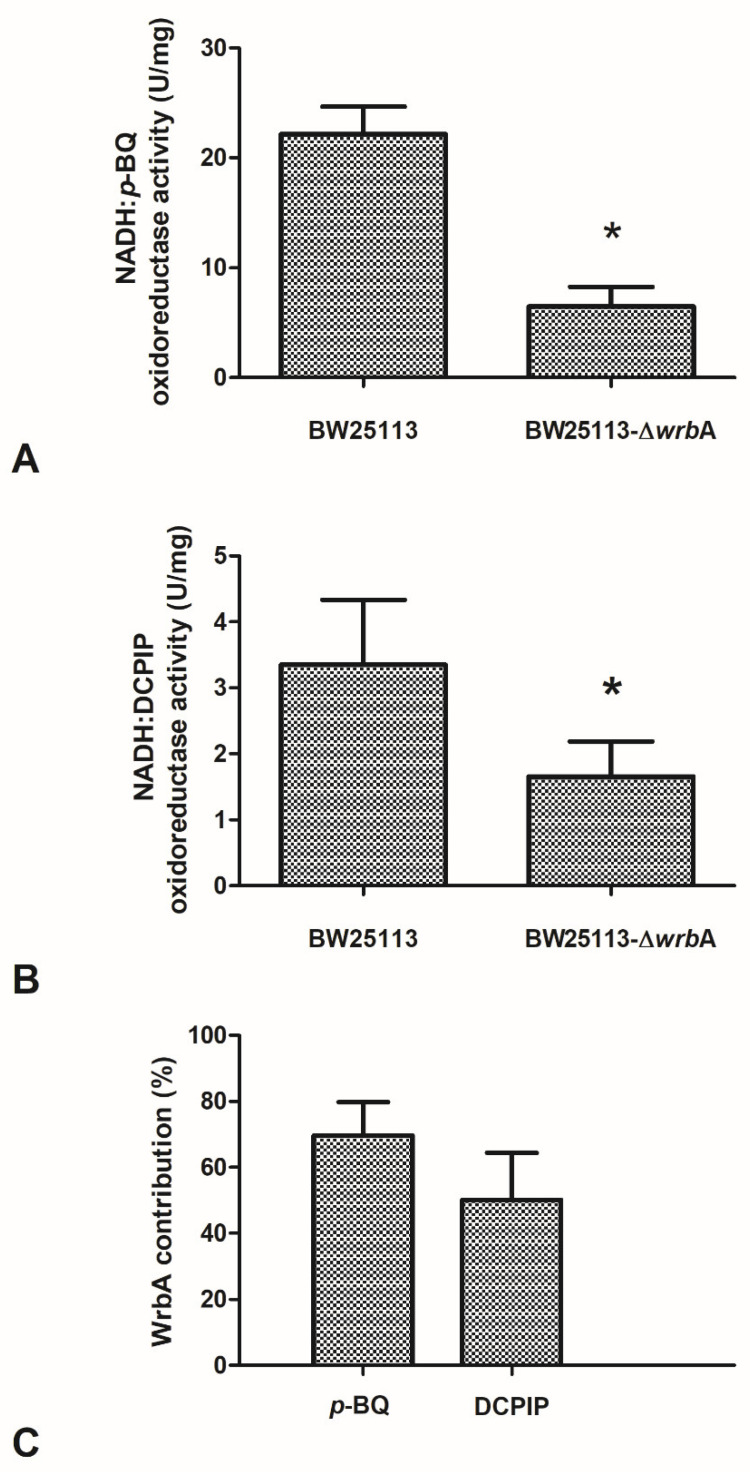
Effects of WrbA deprivation on *E. coli* NADH-dependent oxidoreductase activity and the contribution of WrbA to this activity. (Panels **A**,**B**) Enzyme reactions were carried out at 25 °C in 50 mM Tris-HCl (pH 7.2) in the presence of 0.2 mM NADH, 0.3 mM *p*-benzoquinone (panel **A**), or 0.1 mM 2,6-dichlorophenolindophenol (DCPIP; panel **B**) and a crude extract prepared from cultures of BW25113 or BW25113-∆*wrb*A strains. Initial rates of NADH consumption or DCPIP reduction were measured spectrophotometrically at 340 nm or at 610 nm, respectively. (Panel **C**) WrbA contribution to the total NADH-dependent oxidoreductase activity of *E. coli* (BW25113). The enzyme assays were carried out in the presence of 2,6-dichlorophenolindophenol (DCPIP) or *p*-benzoquinone (p-BQ) as electron acceptors. Data are shown as the means ± standard deviation. The asterisks indicate statistically significant differences (t-test, *p* ≤ 0.05).

**Figure 8 antioxidants-10-00919-f008:**
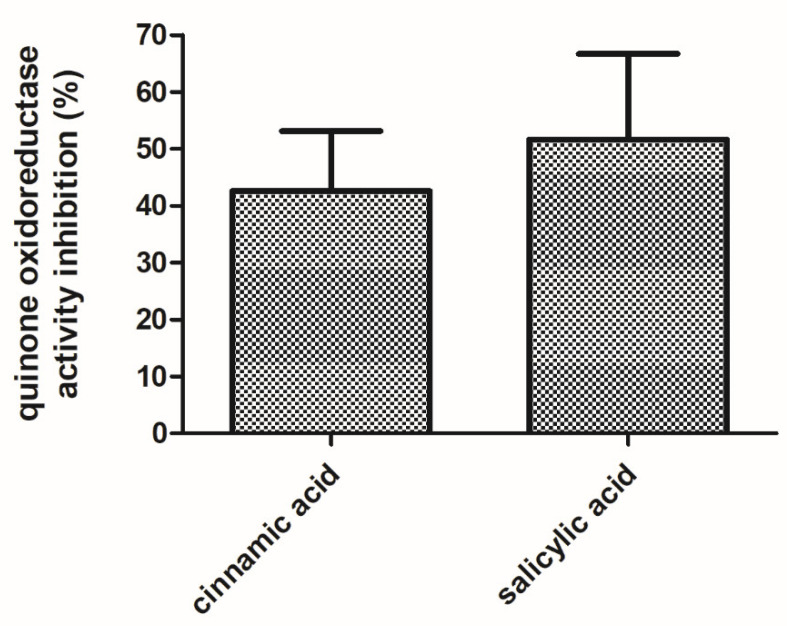
Inhibition effects of the cinnamic and salicylic acids on the function of WrbA. The NADH:*p*-benzoquinone oxidoreductase activity of purified WrbA_JW_ was detected in the absence or presence of the tested antibiofilm molecules via the time-drive monitoring of the NADH absorbance decay at 25 °C. The inhibited activity was percent-related to the uninhibited activity and data are represented as the means with the standard deviation.

**Table 1 antioxidants-10-00919-t001:** Primers used in this work.

Name	Sequence ^a^	Description
U-*wrb*A	TGCGACAAAATTACGTGCTTG	*wrb*A-locus-specific primer (5′ end aligns at the position 144 nt upstream of the start codon of the *wrb*A gene)
D-*wrb*A	GCGACTTCGAAAATGGCCTC	*wrb*A-locus-specific primer (5′ end aligns at the position 115 nt downstream of the stop codon of the *wrb*A gene)
k1	CAGTCATAGCCGAATAGCCT	Kanamycin-resistance-specific primer [42]
k2	CGGTGCCCTGAATGAACTGC	Kanamycin-resistance-specific primer [42]

^a^ The 1067234–1068430 region of the *E. coli* K-12 strain W3110 complete genome [43] (GeneBank acc. n.: AP009048.1) was used as reference for the U-*wrb*A and D-*wrb*A sequences.

## Data Availability

The data presented in this study are available in Appendix A.

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
