# Peer review of "Effects of the Quinone Oxidoreductase WrbA on Escherichia coli Biofilm Formation and Oxidative Stress"

_antioxidants, 2021, doi:10.3390/antiox10060919_

Round 1
Reviewer 1 Report
The Authors amended the manuscript in a proper version.
Consequently, I have accepted the manuscript in its updated form.
Author Response
We thank the Reviewer for the positive comment.
Reviewer 2 Report
Comments to the manuscript ID: Antioxidants-1227738
I disagree with the authors’ statement that the complementation study seems redundant. In contrast, it is necessary for this experimental design. I know that some authors publish their work without carrying out this type of confirmatory experiment. It does not mean other authors should accept it as a standard procedure. I agree with the authors that full complementation is not always achieved, as the complementation, approach is not perfect. It only mimics the natural setting. However, a conclusive statement must show even a partial restoration of a phenotype (wild type).
If published in the current form, this manuscript will mislead the reader. As a reviewer, I cannot allow this. Therefore, I ask the authors to clearly indicate the limitation of their study.
Author Response
As suggested by the Reviewer, the text has been improved as follows:
- (Lines 331-342) “The observation where the lack of WrbA did not impair the planktonic growth suggested that any phenotypical changes to the BW25113-∆wrbA biofilm could be associated with the wrbA deletion. By performing the complementation on the mutant strain, it can be ensured that the observed mutant phenotype is actually from the loss of wrbA and not from secondary mutations that may have occurred during the creation of the mutant strain. A genetic complementation strategy was not undertaken because i) the single-gene deletion was kept in the parental strain and no transduction steps were adopted; ii) the genetic complementation requires additional plasmid sequences and the subsequent use of plasmid-related antibiotic (and the possible promoter inducer) which introduces new condition in the phenotype analysis; iii) a chemical complementation was successful (see section 3.3).”.
- (Lines 657-659) “The subsequent reintroduction of wrbA in the mutant strain should restore the wild-type phenotype, proving that the observed effects in BW25113-∆wrbA biofilms are indeed caused by the loss of the gene of interest.”.

Reviewer 3 Report
The manuscript by Rossi et al describes the the role of the E. coli quinone oxidoreductase WrbA for biofilm formation and oxidative stress. The authors report that the wrbA-deficient strain is more ROS-sensitive (lower peroxide tolerance), is compromised in its ability to form biofilms (reduced biofilm thickness and ECM content, and has increased catalase expression. The study is well performed but lacks focus and some exciting questions remain enigmatic. I have the following comments:
1) Overall, the different paragraphs appear to be somewhat disconnected and the results/discussion section is in parts difficult to follow.
2) One important control that is missing throughout the ms is the complementation of the mutant strain with plasmid-encoded WrpA
3) line 164: the state that the diameter is less than 106 um. What are the other numbers referring to? Did the authors use glass beads of different sizes?
4) Fig 1 is a good control and could be provided as supplements but does not justify a Figure on its own.
5) The most intriguing phenotype is that the wrpA mutant is much better equipped to form biofilms during the fast-growing phase but looses this ability in the maturation phase. What causes this change seems to not yet fully explored (or at least cannot be fully explained with the data provided). The mutant accumulates for ROS during the fast growing phase, while ROS levels are almost deprived in the maturation phase. However, catalase activity does not explain this phenomenon, as the activity is elevated in the mutant in both phase (in fact, catalase activity is significantly higher in the fast growing phase). It is also unclear to me why the 48 time point was chose as opposed to 18h, where the fold change in CFU is substantially higher between WT and wrpA mutant
6) Figure 3: It is unclear why the authors chose two different sets / strain. I assume that one set of the Figure represents the 48 h time point but information is missing both in the M&M and the Figure legend. The scale should be uniform between WT and mutant.
7) Figure 4 & 6 show the ROS level in these biofilms and could be presented in one Figure. This is just one example why I felt the flow of the ms was interrupted. Alternatively, the authors might want to explain what we learn from Figure 6 that has not been shown by the experiments in Figure 4.
8) Paragraphs 3.2 and 3.3 are missing
9) Figure 5: It would be worthwhile to also show the 48h time point, otherwise the phenotype may not be related to the ROS-mediated change in biofilm formation. The Congo red staining could help identify whether the wprA mutant how differences in curli or cellulose production.
10) Figure 7: The authors claim that the production of HPII is increased but that is technically not what they show, which would require e.g. western blot analyses. It may just be that HPII is more active in the mutant.
11) Figure 8: 50 mM glutathione seems unphysiologically high. In fact, it seems to be toxic under some conditions, e.g. the 60 h time point.
12) Figure 10: This part seems to not align well with the rest of the study. The focus of the ms is on biofilm formation and oxidative stress, hence, testing the compounds in vivo would be more logical than pm WrpA enzyme activity.
Round 2
Reviewer 2 Report
The manuscript looks fine - I support acceptance of this paper.
This manuscript is a resubmission of an earlier submission. The following is a list of the peer review reports and author responses from that submission.
Round 1
Reviewer 1 Report
Comments to the manuscript ID: antioxidants-1169054
The research article submitted by Rossi et al. describes the role of a biofilm modulator, WrbA, in the physiology of the mature biofilm, specifically the ability of the biofilm to cope with oxidative stress. The authors first created an isogenic wrbA mutant using the red Lambda recombineering approach, followed by a series of assays aiming at determining the role of WrbA in the oxidative stress response of the mature biofilm.
The study is relevant to the field – the results are very interesting. The manuscript is mostly well written although there are some necessary parts (see below). My main concern lies in the fact that the authors of this study did not show that all the phenotypical changes of the ∆wrbA mutant biofilm are indeed associated with the wrbA deletion. Although they performed the growth assay of the mutant and wild type, it would be a significantly better and finally conclusive to carry out the complementation study. It is well documented that the red Lambda approach leaves a scar in the genome (a small portion of the pCP20 vector), which can easily cause frameshift mutations. The proposed study (complementation) would significantly increase the quality of this study.
Minor comments:
“PCR analysis” line 108-117 is not necessary; it is generally protocol for PCR reactions.
Line 236, “(protein amount, 5-20 ug)”. It seems a great variation in protein concentration. The authors did mention that they used the Bradford assay for the determination of protein concentration, but they did not mention that they used this assay to normalize these samples.
Lines 311-320 not necessary.
Lines 351-352, I suggest not to use the terms “exponential” and “stationary” for the biofilm growth. Looking at Figure 2 this is not the “stationary growth phase” – there is a significant increase of the wild type from 62 h to 72 h. Simply, planktonic terminology cannot be applied to the sessile form of life.
Reviewer 2 Report
The subject is according to the scope of the Journal. The chosen topic is of scientific interest.
I want to point that biofilm is generally represented by extracellular polymeric substances (mainly components as polysaccharides, DNA, proteins, and lipids) and not only by bacterial cells. I do not realize why in this manuscript the Authors discuss about biofilm, when they have only considered CFU coming from “colony biofilms” grown on filter membranes instead of detecting the matrix components. Some bacteria have different grow rates but that are not related with how much biofilm components they produce.
Furthermore, I do not value the word “colony biofilms” that the Authors used to obtain CFU. Maybe, the Authors meant bacterial colonies.
The Authors used the words biofilm and sometimes biofilm biomass and biofilm cells. One time it was reported “biomass produced by biofilm”. Can the Authors standardize their scientific terminology and be more accurate?
Please, reorganize the manuscript giving the appropriate meaning to the biofilm concept.
The use of English style and grammar should be accurate and formal especially in the Introduction and Materials and Methods. I want to highlight the fact that the list is not exhaustive.
The plural form “stresses” is not correct.
Change to “every type of surface”.
Please, use a more formal word instead of “thanks to”.
“Such treatments” Which ones?
“With that in mind” is not very suitable in a scientific publication.
Line 45: In this subject is better the use of “have” or “should have” instead of “would have”
Please, check “Inter alia” and “Error reference source not found.”
Line 55: The meaning of the sentence ending with “were susceptible to specific biocide-free antibiofilm compounds.” is not comprehensible.
What is “Escherichia coli”? A clinical isolate? Reference strain? Bacterial strain?
M&M
What is JW0989 and BW25113? What is pCP20?
The amplified genes appear only in table 1 but they should be in the text.
Could the Authors change “sitting” to a different way?
“Serial dilutions”. Which dilutions? 1:2? 1.10?..
Line 151: Which standard method?